# Education and Income Predict Future Emotional Well-Being of Whites but Not Blacks: A Ten-Year Cohort

**DOI:** 10.3390/brainsci8070122

**Published:** 2018-06-29

**Authors:** Shervin Assari, Brianna Preiser, Marisa Kelly

**Affiliations:** 1Department of Psychology, University of California Los Angeles (UCLA), Los Angeles, CA 90095, USA; 2Department of Psychiatry, University of Michigan, Ann Arbor, MI 48109, USA; preisebj@med.umich.edu (B.P.); mbkelly@umich.edu (M.K.); 3Center for Research on Ethnicity, Culture and Health, School of Public Health, University of Michigan, Ann Arbor, MI 48109, USA

**Keywords:** race, social class, socioeconomic status, positive affect, negative affect

## Abstract

Background: The Minorities’ Diminished Return (MDR) theory is defined as systematically smaller effects of socioeconomic status (SES) on the health and well-being of minority groups when compared to Whites. To extend the existing literature on the MDR theory as applied to the change of mental well-being over time, we investigated Black-White differences in the effects of baseline education and income on subsequent changes in positive and negative affect over a ten-year period. Methods: The Midlife in the United States (MIDUS) is a 10-year longitudinal study of American adults. This analysis followed 3731 adults who were either Whites (*n* = 3596) or Blacks (*n* = 135) for 10 years. Education and income, as measured at baseline and 10 years later, were the independent variables. Negative and positive affect, measured at baseline and over ten years of follow up, were the dependent variables. Covariates were age, gender, and physical health (body mass index, self-rated health, and chronic medical conditions), measured at baseline. Race was the focal moderator. We ran multi-group structural equation modeling in the overall sample, with race defining the groups. Results: High education at baseline was associated with an increase in income over the 10-year follow up period for Whites but not Blacks. An increase in income during the follow up period was associated with an increase in the positive affect over time for Whites but not Blacks. Conclusion: The MDR theory is also relevant to the effects of baseline education attainment on subsequent changes in income and then in turn on positive affect over time. The relative disadvantage of Blacks in comparison to Whites in receiving mental health gains from SES may reflect structural racism and discrimination in the United States. There is a need for additional research on specific societal barriers that minimize Blacks’ mental health gains from their SES resources, such as education and income. There is also a need for policies and programs that help Blacks to leverage their SES resources.

## 1. Background

Positive and negative affect are major components of human emotional experiences. There is a body of research that discusses how socioeconomic status (SES), negative and positive affect, and health are linked [1]. Negative affect impacts all age groups, genders, races, and ethnicities [2] and is central to depression, a debilitating illness which affects 1 in 20 Americans and is the leading cause of disability worldwide [2]. While positive affect is particularly essential for a healthy sense of well-being [3,4,5], negative and positive affect are both under the influence of SES (education and income) [6,7,8]. Negative affect disproportionately affects individuals with low SES (low education and low income) [1,2], while individuals with high SES are more likely to report positive affect [9,10]. However, there might be a relative disadvantage for Blacks when compared to Whites, regarding how SES indicators impact positive and negative affect [11].

The Minorities’ Diminished Return (MDR) theory [12,13] can be defined as smaller protective effects of SES on a wide range of tangible outcomes for Black and other minority populations, when compared to Whites [14]. In line with the MDR theory, education has shown stronger effects in influencing income [15,16], drinking behaviors [17], smoking [18], diet [19], chronic disease [20], body mass index [21], self-rated health [22], and mortality [23,24,25,26] for Whites than for Blacks. Income has also shown stronger effects on impulse control [27], obesity [28], oral health [29], chronic disease [30], and mental well-being [31,32] of Whites than Blacks. Education also generates less income for Blacks than for Whites [15,16].

Research evidence suggests that MDR theory also holds for the effects of SES on affect [20,33,34,35]. Studies have documented an increased risk of depression [20] and suicidality [33] for high SES Blacks. In a national sample, high income Black boys had higher risk of lifetime, 12-month, and past month major depressive disorder (MDD) than their low SES counterparts [34]. In a nationally representative sample of adults, high income Black men had a higher risk of MDD than their low income counterparts [35]. In a 25-year follow up period of a nationally representative sample, most educated Black men showed an increase in depressive symptoms over time, a pattern that could not be seen in Black women, White men, or White women [20]. These findings contradict the mainstream findings that SES translates to positive health outcomes [1].

The MDR theory attributes such unequal gains of equal resources to the qualitative differences that exist in the lives of Whites and minority groups, such as Blacks. Such differences hinder Blacks’ mental health gains, even for those who have invested to enhance their SES and class. As the United States (U.S.) society treats Blacks and other minorities worse than Whites, high SES minority populations, particularly high SES Blacks, do not gain access to the same opportunity structure, and SES does not promote their health conditions as it does for Whites. As a result, high SES Blacks’ everyday lives are heavily affected by continuous prejudice and frequent discrimination [36,37,38]. Such racism at multi-levels deteriorates Blacks’ health gains from the new resources that become available to them [36,39,40].

## 2. Aims

The current study was conducted to examine Black-White differences in the effects of baseline education on change in income and then subsequently change in positive and negative affect over time among American adults. We hypothesized that high education attainment at baseline would be associated with a higher income 10 years later for Whites than Blacks. We also hypothesized that high income at year 10 would be associated with higher positive affect and lower negative affect for Whites but not Blacks.

## 3. Methods

### 3.1. Design

Data used in this longitudinal study came from the first 10 years of follow up of the Midlife in the United States (MIDUS), a longitudinal study of American adults. Data were collected from 1995 to 2004, and the study was carried out by the MacArthur Midlife Research Network (MMRN). MIDUS is a national cohort study of over 7000 American adults that were aged between 25 and 74 years with the primary purpose of understanding psychosocial processes that contribute to age-related decline in physical and mental health over time [41,42,43,44,45]. MIDUS is funded by the National Institute on Aging (NIA).

### 3.2. Ethical Considerations

The MIDUS study protocol was approved by the University of Wisconsin-Madison (UWM) Institutional Review Board (IRB). Written informed consent was received for all MIDUS participants. Participants received monetary incentives for their participation in both Wave 1 and Wave 2 of the study (USD 20 and USD 60 for completions of MIDUS 1 and MIDUS 2 surveys, respectively).

### 3.3. Data Collection

MIDUS data collection used a multimodal strategy that was composed of a telephone interview, a computer-assisted personal interview (CAPI), a computer-assisted telephone interview (CATI), a mailed questionnaire, and a face-to-face interview. First, the study employed an initial 30-min phone interview. This was followed by self-administered questionnaires that were mailed to the participants [41,42,43,44,45].

Wave 1 data collection in MIDUS was conducted in 1995 and 1996. The follow-up data collection in MIDUS was conducted 10 years later in 2004 and 2005. Mailings with an accompanying brochure were sent to all Wave 1 MIDUS participants, in order to remind the participants about their participation and to increase their expectation that an interviewer would contact them for the initial telephone survey in the near future. The telephone survey was completed as part of Wave 1. After a phone interview, which lasted 30 min on average, participants received two mailed self-administered questionnaires [41,42,43,44,45].

### 3.4. Participants and Sampling

To enroll a random sample of adults, MIDUS used random digit dialing (RDD), a sampling technique commonly that is used for telephone surveys. RDD is conducted by generating telephone numbers at random. The sampling frame was a national RDD, which allowed all telephone numbers within the continental United States to be selected. MIDUS oversampled individuals in five cities (because of geographic-specific agenda), resulting in a baseline RDD sample of 4244 individuals [41,42,43,44,45].

### 3.5. Analytical Sample

The analytical sample in this study was 3731 White and Black individuals who completed the 10-year follow up duration. The remaining individuals were excluded due to either their racial category not being of interest in this study (not White or Black) or not completing the 10-year follow up assessment.

### 3.6. Follow-Up Data

From a total number of 7108 individuals who were enrolled at baseline (i.e., individuals who completed the phone survey at MIDUS Wave 1), follow up data were gathered for 4963 individuals (70%) at MIDUS Wave 2 ten years later. As a result, MIDUS sample had a 75% overall retention rate (adjusted for mortality). Major causes for loss to follow up were refusal, inability to be contacted, too ill to be interviewed, or deceased [41,42,43,44,45].

### 3.7. Measures

Demographic variables. Age (years), gender (male, female), and race (Black, White) were collected at baseline (in 1995). Age was treated as a continuous measure. Gender (men = 0 [reference group] and women = 1) and self-identified race (Whites = 0 [reference group], Blacks = 1) were operationalized as dichotomous variables 

Educational Attainment. The main SES indicator in this study was educational attainment, which was measured as: (1) less than high school; (2) high school graduate or equivalent; (3) some college; or, (4) college graduate or more. Education was operationalized as a continuous measure.

Physical Health. Three proxy variables that reflect physical health were included in the current study: body mass index (BMI), self-rated health (SRH), and chronic medical conditions (CMC). SRH was a 10 level variable ranging from 1 (worst) to 10 (best). All of the health measures were conceptualized as continuous variables. While a high score for SRH was indicative of good physical health, a higher score for CMC and BMI was reflective of poor health.

Positive Affect. Using the Mroczek and Kolarz (1998) scale [46], positive affect during the past 30 days was measured, using the following feelings: “cheerful”, “in good spirits”, “extremely happy”, “calm and peaceful”, “satisfied”, and “full of life”. Responses were on a Likert scale ranging from 1 (all of the time) to 5 (none of the time) [46]. Mean positive affect scores were computed, with possible scores ranging from 1 to 5. Higher scores reflected more positive affect. Internal consistency (reliability) was very good (α = 0.91 for all, 0.91 for Whites, 0.92 for Blacks) [47,48,49].

Negative Affect. Using the same measure by Mroczek and Kolarz (1998) scale [46], negative affect during the past 30 days was measured using the following feelings: “so sad”, “nervous”, “restless or fidgety”, “hopeless”, “worthless”, and “everything was an effort”. Response items were on a Likert scale, ranging from 1 (all of the time) to 5 (none of the time) [46]. An average score was calculated that reflected negative affect, with scores ranging from 1 to 5. [47,48,49]. Higher scores were reflective of more negative affect. Internal consistency (reliability measure) was high for all (α = 0.86), for Whites (α = 0.86), and for Blacks (α = 0.87). This measure is widely used to assess affect [50,51].

### 3.8. Statistical Analysis

SPSS 22.0 (SPSS Inc., Chicago, IL, USA) and AMOS 22.0 [52,53] were used to conduct the data analysis. Frequency (%) and mean (SD) were reported to describe the sample at the baseline and 10 years later. Pearson’s correlation was used to calculate the bivariate correlations in the overall sample.

A multi-group structural equation model (SEM) was used for multivariable analysis [54]. In our models, the groups were defined based on race. Education and income measured at baseline were the independent variables. Negative and positive affect measured at baseline and over ten years of follow up were the dependent variables. Covariates included age, gender, and health (body mass index, self-rated health, and chronic medical conditions) measured at baseline. Income measured at ten years of follow up was the mediator. Race was the focal moderator. To handle the missing data, Full Information Maximum Likelihood (FIML) was used. The final SEM model did not include any constrains or co-variances for the errors.

The model fit was assessed using the conventional fit statistics that included a non-significant chi-square test (*p* > 0.05), a comparative fit index (CFI) larger than 0.95, a root mean squared error of approximation (RMSEA) of less than 0.06, and an *X*^2^ to degrees of freedom ratio of less than 4 [55,56,57,58]. We reported standardized regression coefficients, with associated standard errors (SE) and *p* values for each path.

## 4. Results

### 4.1. Descriptive Statistics

This study included 3731 adults who were either Whites (*n* = 3596) or Blacks (*n* = 135) for 10 years.

Table 1 provides a summary of the descriptive statistics for the overall sample, as well as for racial groups. Blacks had lower SES (education and income) than Whites. Blacks also had higher negative affect at baseline and 10 years later than Whites (Table 1).

### 4.2. Bivariate Correlations

Table 2 summarizes the results of bivariate correlations. Education and income were associated with positive and negative emotions at baseline and 10 years later in the overall sample (Table 2).

### 4.3. Multivariable Models

Our SEM showed very good fit. CMIN = 24.465; DF = 8; *p* = 0.002; CMIN/DF = 3.058; CFI = 0.998; RMSEA = 0.023 (90% CI = 0.013–0.034). Table 3 summarizes the path coefficients for the SEM. Figure 1a,b also show these paths for Whites and Blacks. As these models show, baseline education showed an effect on change in income over the next 10 years for Whites but not Blacks. Change in income, in turn, predicted an increase in positive emotions for Whites but not Blacks (Table 3).

## 5. Discussion

The results showed an effect of high education attainment at baseline on an increase in income over the 10-year follow up period for Whites but not for Blacks. The study also showed an association between an increase in income and an increase in the positive affect for Whites but not Blacks. These results indicate that the MDR theory also holds for economic and mental health return of education attainment over time.

Our findings support the MDR theory [12,13,21], defined as the systemically smaller economic and health effects of same SES indicators for Blacks and other minorities when compared to Whites [21,24,31,33,36,59,60,61,62,63,64]. Two studies in particular have shown that education generates more economic return for Whites than Blacks [15,16]. The impact of educational attainment on changing drinking patterns [17], BMI, insomnia, physical activity [17], depression [20], suicidal behaviors [33], and mortality [24] are all shown to be smaller for Blacks than for Whites. Similar results are seen in the transgenerational effects of parental education on child outcomes [21,22,27,65].

The results also support Link and Phelan’s (1995) Fundamental Cause Theory, suggesting that SES is a fundamental and root cause of a wide range of outcomes, including mental health [66,67,68]. Link and Phelan also introduce racism as a fundamental cause [69]. Our findings are also in line with the life course epidemiology approach, suggesting that risk factors and resources have long-term effects on population health decades later [70,71,72,73].

Not only do Blacks gain less than do Whites from SES, high SES may operate as a risk factor for poor mental health outcomes for Blacks, particularly in Black males [14,20,21,33,34,39,60,74,75,76]. To provide examples, high education and income are shown to be risk factors of MDD, symptoms of depression, and poor self-rated mental health for Black youth and adults, particularly males [32,34]. It is not clear why we could not replicate MDR theory for negative affect in the current sample.

The findings reported here should not be interpreted as Blacks are less capable of turning their SES resources to tangible outcomes. This interpretation would be blaming the victim of a system that oppresses them. Minorities’ Diminished Return is not a function of minorities’ culture or laziness, but a consequence of legacy of slavery and remaining systemic racism. Racism is still a core element of the social structure and function in the U.S. Across institutions and levels of society, there are deeply rooted inequalities that hinder Blacks’ progress, and reduce their ability to transform their resources into tangible gains [77,78,79]. As long as race and skin color strongly influence how individuals and groups are treated in our society and how people access opportunity structure, true equality between races is not achievable. Without a drastic change to the structure and function of the U.S. system, America will continue to fail the middle-class and high SES Blacks who have paid the cost to climb the social ladder. Society is charging Blacks and other racial and ethnic minority groups an extra cost for upward social mobility, which diminishes the physical and mental health status of Blacks who have successfully climbed the social ladder [39,60,80,81,82,83,84]. Historically, economic and public policies in the U.S. were designed and implemented to maximize the gain of the majority (Whites) even to the cost of ignoring specific needs of marginalized social groups. For example, U.S. culture overemphasizes the individual’s responsibility and bootstrapping, and it is systemically against safety nets, welfare, and universal coverage of health. High aspirations, motivations, and ambitions do not increase health outcomes for Blacks and other minority groups as compared to Whites. That is, educational attainment does not generate economic rewards such as prestigious and high paying jobs and wealth. Education and class that serve Whites, particularly White men, do not change purchasing power and wealth for Blacks, particularly Black men [85,86,87].

### 5.1. Implications for Policies and Programs

There is a need to reduce qualitative differences in the lives of Blacks and Whites so SES can generate similar economic and mental health gain, regardless of race. Without a systemic change, minorities will continue to remain at a systemic disadvantage relative to Whites. As mentioned before [12,13], policy solutions must not be limited to equalizing access and merely enhancing minorities’ SES. Instead, policies must address societal and structural barriers that disproportionately limit minorities’ abilities to translate their resources to health. Policy makers should be aware that some universal policies may have a larger impact on Whites than minorities, and may have the unintended effect of increasing the racial gap in health. For such high risk policies, evaluations are needed to explore racial variation in the reach and impact of the policy, and to ensure that minority groups are not left behind in gaining from the resources.

These results suggest that we need to reduce discrimination and inequalities in the education system, as well as in the labor market. There is a need to increase investment in quality of education of inner cities and in minority-majority schools. Policies should be in place to reduce both the discrimination of Black students in school and the differential hiring chances of racial groups. The results are also in support of affirmative action policies, simply because educational attainment is not enough for equalizing the well-being of Blacks and Whites. Blacks and other minority groups may require additional assistance leveraging their available SES resources, like education and employment. Policies should not permit education, which is a potential equalizer, to become a source of inequalities in employment, income, and life conditions across groups.

### 5.2. Limitations and Future Directions for Research

Our study had a few limitations. One main weakness includes the small sample size for Blacks (*n* = 135) as compared to Whites (*n* = 3596), which limits the power for statistically detecting the significant main effects for education or potential effect modification by gender. The inability to examine interactions by gender, for Blacks, is a major limitation since the literature has shown SES to be a risk factor for poor mental health of Black men. SES may impact mental health of Black males and Black females differently. Second, educational attainment was conceptualized and operationalized as a fixed variable. However, similar to other psychosocial constructs, educational attainment is subject to change over a 10-year period. Third, our study missed to control for all the potential confounders and covariates. Future research should test if the effects of other SES indicators, such as income, marital status, occupation, employment, home ownership, and wealth are also different for Whites and Blacks. This study neither measured multi-generational transition of SES nor childhood SES, which should be examined in future research on MDR. In addition, the SES level is not limited to the individual level, but to the family and community levels as well. Available resources across multiple levels, within the individual, in the family, and in the neighborhood and community should also be investigated in the future research. Contextual factors other than SES (e.g., neighborhood racial composition) are needed to be included as explanatory mechanisms that reduce Blacks’ health gains from individual-level SES. Research is needed on the role of quality of sleep and associated suicidality [88]. This is particularly important given the role of sleep on affect [89], and also given the racial differences in the protective effects of SES on sleep across racial groups [36,90]. Future research should go beyond the narrow definition of minority status based on race and it should consider the role of other marginalized identities. Research should also test the role of intersectionality of race, ethnicity, gender, sexual orientation, nativity, place, and class. Despite these limitations, the current study makes a unique contribution to the literature on the MDR theory [13,21,60], and shows that it also applies to the effects of educational attainment on changes in positive affect over ten years. Some strengths of our study included recruitment of a nationally representative sample, a large sample size, a longitudinal study design, and a 10-year follow up.

## 6. Conclusions

The magnitude of the effect of baseline education on changing future income and its subsequent impact on increasing positive affect are not equal across Black and White Americans. An inequality exists in the economic and emotional return of educational attainment over time. Similar resources, like education and income, consistently result in lower economic and mental health gains for Black Americans when compared to those of White Americans. Multi-level solutions should be comprehensive and include policy solutions that go beyond merely equalizing the access of populations to SES resources and eliminate the inequality in societal barriers in the lives of minority populations. It is only then that racial minority groups can achieve comparable outcomes to the majority group in response to similar access to resources.

## Figures and Tables

**Figure 1 brainsci-08-00122-f001:**
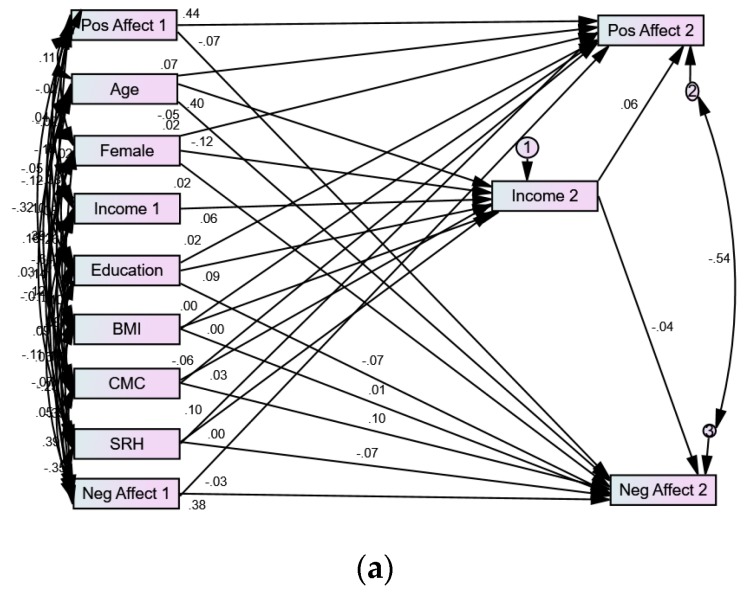
(**a**) Summary of path coefficients in Whites. (**b**) Summary of path coefficients in Blacks. Self-Rated Health (SRH), Chronic Medical Conditions (CMC), Body Mass Index (BMI).

**Table 1 brainsci-08-00122-t001:** Descriptive Statistics.

	All *n* = 3731		Whites *n* = 3596		Blacks *n* = 135	
	Mean	SD	Mean	SD	Mean	SD
Age	47.41	12.39	47.44	12.40	46.57	12.25
Income1 (Personal) *	27,511.15	27,372.27	27,686.58	27,563.13	22,810.08	21,187.33
Income 2 (household) *	42,525.09	40,669.14	42,661.16	40,868.10	38,423.42	34,032.02
Self-Rated Health (SRH)*	7.58	1.50	7.57	1.49	7.84	1.72
Chronic Medical Conditions (CMC)*	2.31	2.35	2.31	2.34	2.49	2.66
Body Mass Index (BMI)*	26.69	5.19	26.59	5.09	29.35	6.93
Positive Affect 1 *	3.41	0.71	3.41	0.71	3.55	0.75
Positive Affect 2	3.43	0.70	3.43	0.70	3.55	0.79
Negative Affect 1	1.50	0.58	1.50	0.58	1.55	0.76
Negative Affect 2 *	1.50	0.57	1.50	0.56	1.65	0.82

* *p* < 0.05 for comparison of Blacks and Whites; Independent sample *t* test.

**Table 2 brainsci-08-00122-t002:** Summarizes the results of bivariate correlations. Education and income were associated with positive and negative affect at baseline and 10 years later in the overall sample. (Table 2).

	**1**	**2**	**3**	**4**	**5**	**6**	**7**	**8**	**9**	**10**	**11**	**12**	**13**
Race (Black)	1	0.05 **	−0.01	−0.04 *	−0.03 *	0.03	0.03 *	0.01	0.10 **	0.04	0.03 *	0.02	0.05 **
Gender (Female)		1	−0.02	−0.09 **	−0.38 **	−0.16 **	−0.01	0.14 **	−0.10 **	−0.03	−0.01	0.09 **	0.09 **
Age			1	−0.11 **	−0.12 **	0.39 **	0.03 *	0.16 **	0.01 **	0.11 **	0.14 **	−0.12 **	−0.10 **
Education				1	0.26 **	0.06 **	0.05 **	−0.11 **	−0.10 **	0.02	0.03	−0.07 **	−0.11 **
Income 1					1	0.08 **	0.09 **	−0.17 **	0.01	0.04 *	0.07 **	−0.11 **	−0.15 **
Income 2						1	0.01	0.06 **	0.04 *	0.05 **	0.10 **	−0.08 **	−0.10 **
Self-Rated Health (SRH)							1	−0.39 **	−0.25 **	0.39 **	0.32 **	−0.34 **	−0.27 **
Chronic Medical Conditions (CMC)								1	0.17 **	−0.31 **	−0.23 **	0.39 **	0.28 **
Body Mass Index (BMI)									1	−0.05 **	−0.06 **	0.05 **	0.08 **
Positive Affect 1										1	0.53 **	−0.63 **	−0.37 **
Positive Affect 2											1	−0.38 **	−0.61 **
Negative Affect 1												1	0.50 **
Negative Affect 2													1
	**1**	**2**	**3**	**4**	**5**	**6**	**7**	**8**	**9**	**10**	**11**	**12**	**13**
Race (Black)	1	0.05 **	−0.01	−0.04 *	−0.03 *	0.03	0.03 *	0.01	0.10 **	0.04	0.03 *	0.02	0.05 **
Gender (Female)		1	−0.02	−0.09 **	−0.38 **	−0.16 **	−0.01	0.14 **	−0.10 **	−0.03	−0.01	0.09 **	0.09 **
Age			1	−0.11 **	−0.12 **	0.39 **	0.03 *	0.16 **	0.01 **	0.11 **	0.14 **	−0.12 **	−0.10 **
Education				1	0.26 **	0.06 **	0.05 **	−0.11 **	−0.10 **	0.02	0.03	−0.07 **	−0.11 **
Income 1					1	0.08 **	0.09 **	−0.17 **	0.01	0.04 *	0.07 **	−0.11 **	−0.15 **
Income 2						1	0.01	0.06 **	0.04 *	0.05 **	0.10 **	−0.08 **	−0.10 **
Self-Rated Health (SRH)							1	−0.39 **	−0.25 **	0.39 **	0.32 **	−0.34 **	−0.27 **
Chronic Medical Conditions (CMC)								1	0.17 **	−0.31 **	−0.23 **	0.39 **	0.28 **
Body Mass Index (BMI)									1	−0.05 **	−0.06 **	0.05 **	0.08 **
Positive Affect 1										1	0.53 **	−0.63 **	−0.37 **
Positive Affect 2											1	−0.38 **	−0.61 **
Negative Affect 1												1	0.50 **
Negative Affect 2													1

* *p* < 0.05, ** *p* < 0.01.

**Table 3 brainsci-08-00122-t003:** Summary of linear regression models in the overall sample and across races.

		Whites			Blacks		
		Estimate	S.E.	*p*	Estimate	S.E.	*p*
Education	Income 2	0.09	0.22	<0.001	0.03	1.50	0.765
Gender	Income 2	−0.12	0.22	<0.001	−0.05	1.61	0.615
Income 1	Income 2	0.06	0.00	0.001	−0.08	0.00	0.467
Age	Income 2	0.40	0.01	<0.001	0.47	0.06	<0.001
Self-Rated Health (SRH)	Income 2	−0.01	0.08	0.814	0.16	0.46	0.131
Chronic Medical Conditions (CMC)	Income 2	0.03	0.05	0.183	0.15	0.29	0.151
Body Mass Index (BMI)	Income 2	0.00	0.02	0.866	−0.02	0.11	0.836
Positive Affect 1	Positive Affect 2	0.44	0.02	<0.001	0.44	0.09	<0.001
Negative Affect 1	Negative Affect 2	0.39	0.02	<0.001	0.37	0.10	<0.001
Education	Positive Affect 2	0.02	0.02	0.25	−0.05	0.11	0.44
Income 2	Positive Affect 2	0.06	0.00	<0.001	−0.01	0.01	0.962
Income 2	Negative Affect 2	−0.04	0.00	0.014	−0.03	0.01	0.817
Age	Negative Affect 2	−0.05	0.00	0.003	−0.12	0.01	0.190
Age	Positive Affect 2	0.07	0.00	<0.001	0.02	0.01	0.787
Gender	Negative Affect 2	0.02	0.02	0.156	0.02	0.14	0.769
Gender	Positive Affect 2	0.02	0.02	0.173	0.06	0.12	0.414
Education	Negative Affect 2	−0.07	0.02	<0.001	−0.11	0.13	0.172
Negative Affect 1	Positive Affect 2	−0.04	0.02	0.066	−0.06	0.08	0.469
Body Mass Index (BMI)	Positive Affect 2	0.00	0.00	0.892	−0.05	0.01	0.518
Body Mass Index (BMI)	Negative Affect 2	0.01	0.00	0.572	0.17	0.01	0.036
Chronic Medical Conditions (CMC)	Positive Affect 2	−0.06	0.01	<0.001	0.18	0.02	0.016
Chronic Medical Conditions (CMC)	Negative Affect 2	0.10	0.00	<0.001	−0.1	0.03	0.225
Self-Rated Health (SRH)	Positive Affect 2	0.10	0.01	<0.001	0.32	0.04	<0.001
Self-Rated Health (SRH)	Negative Affect 2	−0.07	0.01	<0.001	−0.11	0.04	0.236
Positive Affect 1	Negative Affect 2	−0.07	0.02	<0.001	0.02	0.10	0.785

Our SEM showed very good fit. CMIN = 24.465; DF = 8; *p* = 0.002; CMIN/DF = 3.058; CFI = 0.998; RMSEA = 0.023 (90% CI = 0.013–0.034).

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
