# Peer review of "Education and Income Predict Future Emotional Well-Being of Whites but Not Blacks: A Ten-Year Cohort"

_brainsci, 2018, doi:10.3390/brainsci8070122_

Round 1

Reviewer 1 Report

This article addresses one of the most important educational and cultural issues in American society, and one which ever requires deeper understanding. 

From a sociocultural standpoint, given the current cultural climate and policy challenges in the USA, this paper lends much needed credence to the importance of careful and systematic study of the many factors that contribute to race-based disparity. Additionally, educational research is often narrow and attentive disproportionately to achievement. This study, however, extends and deepens the frame of educational research by looking at the important contributions that emotion and affect make; a notion often dismissed when examining education from a neoliberal perspective. 

From a social scientific standpoint, the methodology and data analytic procedures are highly sound and appropriate. Additionally, the contextualization of the study as longitudinal contributes to its soundness, and is able to demonstrate how these factors function over time in a formative, rather than summative, manner. The statistical analysis is thorough and comprehensive. This is a highly valuable piece to the field of education, and the conversation surrounding sociocultural issues as a whole. 

Author Response

This comment is much appreciated. We are very thankful.

Reviewer 2 Report

The authors presented an interesting paper aimed to explore Black-White differences in the effects of baseline education and income on subsequent changes in positive and negative affect over a ten-year period in a sample of 3731 adults. The authors reported that the higher education at baseline was linked to an increased income over the 10-year follow up period for Whites but not Blacks. An increase in income during the follow up period was even associated with an increase in positive affect over time for Whites but not Blacks.

The authors may find as follows my main comments/suggestions.

1) When throughout the Introduction section, the authors reported that some studies documented an increased depression and suicidality risk for higher socioeconomic status Blacks, more details/information are requested to this regard. Here, specifically, the most relevant evidence supporting this notion should be added to the main text.

2) Whether higher socioeconomic status even influences the emergence of hopelessness and/or further additional risk factors for suicide (e.g., sleep disturbances that are even associated with the enhanced suicide risk) needs to be specified. The link between higher socioeconomic status and specific depressive symptoms such as insomnia is particularly interesting. Specifically, research demonstrated that patients with sleep alterations, particularly insomnia, are at an increased risk of experiencing suicidal ideation and/or making a suicide attempt. Although the link between insomnia and suicidal behavior is not the main topic of the present paper, the authors might refer, at least briefly, to the mentioned topic citing the paper which has been published on International Journal of Clinical Practice (67(12):1311-1316) in 2013.

3) Given that the main aims of this paper have been extensively proposed by the authors, the specific hypotheses underlying the study objectives should be adequately reported as well.

4) The most relevant reasons related to the 75% overall retention rate in the MIDUS sample need to be better explained for the general readership. Conversely, the authors simply reported that major causes for loss to follow up were refusal, could not be contacted, too ill to be interviewed, or deceased.

5)  The most relevant psychometric instruments used in the present study might be reported in a more succinct manner within the main text.

6) The authors do not need to repeat the main objectives of this study in the first lines of the Discussion section, as these aims have been already reported elsewhere. Here, i suggest to immediately present the relevance and most important implications of the main findings.

7) When the authors reported that one of the implications for this study is to reduce disparities/inequalities in the education system, they should specify, in a more detailed manner, how they would like to proceed in this direction. In particular, which type of education strategies may they hypothesize in the clinical practice to  reduce disparities/inequalities in the mentioned sample?

8) The manuscript needs to be reviewed by a native English speaker for the quality of language.

Author Response

Thank you for your exceptionally helpful comments. Using these comments, the paper has been improved.

These are the main changes based on your comments:

1) Throughout the Introduction section, when we report that some studies documented an increased depression and suicidality risk for higher socioeconomic status Blacks, more details/information are given.

2) Although the link between insomnia and suicidal behavior was not the main topic of the present paper, we cited the following paper that helps bring sleep to the picture: International Journal of Clinical Practice (67(12):1311-1316) in 2013.

3) We listed two specific hypotheses after the aims in the introduction.

4) Cohort studies d not commonly measure reasons for retention (why should they?). The studies try to measure why people drop out. (Serves their purpose). No change is made based on this comment.

5)  We have provided relevant psychometric properties and information on our instruments in the methods section. Cronbach's alpha is reported as well.

6) OK, we are no more repeating the main objectives of this study in the first lines of the Discussion section. Immediately, we present the relevance and most important implications of the main findings.

7) The main implications for this study are for public policy rather than clinical practice. We are proposing changes to reduce discrimination in education system and  labor market. Please see the changes.

8) The manuscript is now reviewed and edited by a native English speaker. This is the only change that is not highlighted.

Round 2

Reviewer 2 Report

In the revised manuscript, the authors have now addressed most of the major comments raised by Reviewers improving both the main structure and quality of this paper. I have no further additional questions.